# Effective Treatments of UTI—Is Intravesical Therapy the Future?

**DOI:** 10.3390/pathogens12030417

**Published:** 2023-03-06

**Authors:** Chris J. Morris, Jennifer L. Rohn, Scott Glickman, Kylie J. Mansfield

**Affiliations:** 1School of Pharmacy, University College London, 29-39 Brunswick Square, London WC1N 1AX, UK; 2Division of Medicine, University College London, Royal Free Hospital Campus, Rowland Hill Street, London NW3 2PF, UK; 3UroPharma Ltd., Norwich NR1 1PY, UK; 4Graduate School of Medicine, University of Wollongong, Wollongong, NSW 2522, Australia

**Keywords:** urinary tract infection, intravesical, antibiotic, urothelium

## Abstract

Urinary tract infection (UTI) afflicts millions of patients globally each year. While the majority of UTIs are successfully treated with orally administered antibiotics, the impact of oral antibiotics on the host microbiota is under close research scrutiny and the potential for dysbiosis is a cause for concern. Optimal treatment of UTI relies upon the selection of an agent which displays appropriate pharmacokinetic-pharmacodynamic (PK-PD) properties that will deliver appropriately high concentrations in the urinary tract after oral administration. Alternatively, high local concentrations of antibiotic at the urothelial surface can be achieved by direct instillation into the urinary tract. For antibiotics with the appropriate physicochemical properties, this can be of critical importance in cases for which an intracellular urothelial bacterial reservoir is suspected. In this review, we summarise the underpinning biopharmaceutical barriers to effective treatment of UTI and provide an overview of the evidence for the deployment of the intravesical administration route for antibiotics.

## 1. Introduction

UTIs remain one of the most common bacterial infections, affecting 150 million patients globally each year [1]. The main risk factors for uncomplicated UTI include age, diabetes mellitus, genetic susceptibility, and sexual intercourse [2]. Although both men and women may develop UTI, women are more likely to experience UTI and 50% of all women will be affected across their lifespan [3,4]. UTIs are expected to impact one out of every two women by the time they are 30, with 30% of those women experiencing a recurrence of infection within six months, regardless of antibiotic treatment [5].

UTI is also commonly associated with the use of urinary drainage catheters. Globally, catheters are a very common and increasingly used clinical tool. A National Health Service England survey from across the entire system of its users found 12.9% of patients were catheterised [6]. Similarly, a Dutch survey revealed that over a 21-year period (1997–2018), indwelling catheter use per 100,000 population nearly doubled and intermittent catheter usage nearly trebled [7]. In the US, America’s Medicare & Medicaid reimburse up to 200 intermittent catheters/month/per person, encouraging single use only to minimise infections [8].

## 2. The Most Common Treatment for UTI Is Antibiotics

UTI, alongside respiratory infection, is among the most common reasons why antibiotics are prescribed in primary care [9]. However, it is generally appreciated that the long-term and uncontrolled use of oral/systemic antibiotics has led to the emergence of multidrug-resistant microorganisms in recent years, which is a major concern due to limited availability of other treatment options for UTI [10]. Systemic antibiotic treatment can obviously select for antibiotic-resistant strains, thereby fuelling the global antimicrobial resistance crisis. The ‘golden era of antibiotics’ has long since come to an end, and dangerously high levels of antibiotic resistance continue to spread globally [10]. The World Health Organisation declared antimicrobial resistance as one of the top 10 global public health threats facing humanity [11], therefore the demand for rationally designed and alternative treatments of UTI is paramount.

While antibiotics can be effective in treating acute UTI in three quarters of patients, their use has detrimental effect on the homeostasis of the host microbiota when it is exposed directly to these drugs, which in turn adversely harms human health via effects on nutrients, metabolism, pathogen resistance, and other processes [12]. Recent advances in genomic sequencing technologies and analytical methods have seen a renaissance in our understanding of the gut microbiota—the collective consortium of microorganisms that reside in a healthy gastrointestinal tract [13,14]. Given the importance of the microbiota, it is no surprise that its disruption by systemic antibiotic treatment can lead to negative effects on health [15]. Imbalances in gut microbiota caused by use of antibiotics have been linked to obesity, allergy and atopic disorders, autoimmune diseases, such as type 1 diabetes, rheumatoid arthritis, and multiple sclerosis, along with various infectious diseases [12,16]. It is well known that prolonged antibiotic treatment can initiate a state of dysbiosis that favours the outgrowth of a dangerous pathogen, *Clostridioides difficile*. Antibiotic use is also associated with a reduction in gut microbiota species diversity, which can be detrimental, for example by perturbing the complex network of interconnected metabolic activity over which the microbial community presides [15].

Less well understood is the effect of antibiotic treatment in the bladder itself. Although long assumed to be sterile, a number of years of careful work by multiple labs have established that the healthy bladder, which is open to the environment, contains a commensal ‘urobiome’ that differs significantly from the infected state [17,18,19,20,21,22]. While these resident microorganisms are less well defined that those of other niches such as the gut, and the consequences of their disruption remain obscure, it is possible that antibiotics may also impair a protective milieu in the bladder itself. Indeed, as indirect evidence for this notion, a number of clinical studies have shown that bladder instillation of the asymptomatic strain *Escherichia coli* 83972 can protect recurrent UTI patients from infection {Loubet, 2020 #1808}. Moreover, as far back as 1997, Smith and colleagues noted that prior antibiotic treatment (either for a UTI or some other infection) in young women actually increased the risk of cystitis (bladder infection) [23]. Costelloe and colleagues showed that prescribing antibiotics for UTI patients in primary care was associated with an increased risk of AMR uropathogens, a risk which remained for at least a year after the prescription had been dispensed [24]. Clinically, this risks treatment failures, recurrent UTI episodes, and AMR spread within the population.

Antibiotic choice in UTI treatment is largely empirical, with the consistent aim of either killing or stalling the growth of the causative uropathogen. For over six decades it has been appreciated that in the treatment of kidney infection (pyelonephritis), antimicrobial activity against the infecting organism in the urine during the course of treatment was essential for successful eradication of bacteriuria [25]. However, the intricacies of antibiotic pharmacokinetics/pharmacodynamics (PK/PD) have only been elaborated in recent decades. It is now a known imperative to consider not only the antibiotic susceptibility profile for the suspected pathogen, but also the likelihood that the antibiotic of choice will be present at sufficient concentrations in the target tissue for sufficient time to resolve the infection. In the context of UTI, it was long ago noted that favourable clinical outcomes in UTI cases correlated better with antibiotic concentrations in the urinary system rather than those in serum [26]. Without proper consideration of the PK/PD, it is possible that, irrespective of its potent anti-bacterial activity, the drug of choice would fail without the use of an appropriate dosing schedule that delivers drug levels above the minimum inhibitory concentration (MIC) at the infection site for an appropriate time course. This is compounded by the fact that bacterial pathogens in their urothelial niche may harbour resistant phenotypic traits which increase the antibiotic concentrations required for effective treatment. In recurrent UTI, the infection site can be located intracellularly and thus present additional dispositional challenges. Critically, the most pernicious sequalae of ineffective antibiotic therapy is the development of AMR and its chronic effects on patient morbidity and mortality. Therefore, it is critical that the dosing schedule and route of administration be carefully rationalised for optimal treatment outcomes.

## 3. Antibiotic Drug Disposition

The physiological role of the urinary system is to filter the blood, retrieving key metabolic substrates while selectively disposing of metabolic by-products and other solutes that may otherwise pose a toxicity risk to the body. Together with the liver, the renal system serves as the major clearance route for metabolic by-products and xenobiotics. Following administration of any drug, including antibiotics, there is potential for metabolism in the liver. Hepatic metabolic transformations typically produce metabolites that are more polar in nature that the parent antibiotic, which can be achieved through direct chemical modification of the antibiotic structure through Phase I metabolism (e.g., oxidation, reduction, hydrolysis). These chemical transformations are often coupled with metabolite conjugation to polar moieties such as glucuronic acids and sulfates in a process termed Phase II metabolism. Ultimately, the purpose of Phase I and II metabolic processes is to endow the drug with physicochemical features (polarity, hydrophilicity) that make it liable to renal excretion.

Many antibiotics, including those indicated for UTI, are principally cleared in their chemically unmodified, active form via the renal route. This results in high antibiotic concentrations in the urine, which often surpass those detected in the blood plasma. Therefore, optimised pharmacotherapy of UTIs can be achieved by careful selection of the antibiotic agents according to their pharmacokinetics as well as their antibacterial activity against the causative organism.

Antibiotic efficacy is principally dependent upon the relative free concentration in the relevant tissue fluid in relation to the minimum inhibitory concentration (MIC) against the pathogen in question. Consider the concentration–time profile of a typical antibiotic after oral administration (Figure 1).

When considering the optimal therapy for UTI, one should consider the PK/PD of the drugs under selection. To avoid the emergence of AMR, it is imperative that the dose magnitude and dosing schedule are appropriate to achieve drug concentrations above the MIC for the suspected pathogen. There are three basic PK/PD parameters that correlate well with antimicrobial efficacy in vitro and in vivo (Figure 1).

(1)The length of time for which the antibiotic concentration surpasses the MIC: T > MIC(2)The ratio between the peak antibiotic concentration (C_max_) and the MIC: C_max_/MIC(3)The ratio between the under the plasma concentration–time curve (AUC) and the MIC: AUC/MIC

For the majority of antibiotics used to treat UTIs, antibacterial efficacy is time-dependent and therefore the optimal PK/PD parameter is T > MIC. In contrast, aminoglycosides, which are delivered exclusively by parenteral routes for severe UTIs, display concentration-dependent activity and are therefore best assessed by the parameter, C_max_/MIC. Other antibiotics, such as the fluoroquinolones, which are typically reserved for more severe/resistant infections, display concentration-dependent activity and their efficacy is well predicted by AUC/MIC [27]. Notably, some antibiotics are associated with a continued suppression of bacterial growth following withdrawal of the antibiotic. This “post-antibiotic effect” (PAE) has been recorded for several antibiotics including gentamicin and ciprofoxacin [28].

One can estimate the time course of antibiotic appearance in the primary urine by studying the drug plasma concentration–time profile. However, the dynamic processes of renal filtration, bladder filling, and voiding render the precise prediction of antibiotic concentrations in the urinary tract extremely challenging. A comprehensive understanding of the PK/PD of antibiotics active against UTI can therefore be appreciated fully only after direct measurement of urinary antibiotic concentrations.

Detailed published data comparing plasma and urinary concentrations of long-established antibiotics are relatively scant. Nonetheless, for some of the well-established, first-line antibiotics for UTI, there are comparative plasma and urinary PK profiles available (reviewed in [29]). Figure 2 shows a significant disparity between the ciprofloxacin levels in the plasma and urine samples of healthy volunteers after oral administration, with plasma and urinary C_max_ values of 2.2 and 268 mg/L, respectively, representing approximately 100× higher ciprofloxacin concentrations in urine. The ratios of urinary and plasma antibiotic concentrations for some antibiotics commonly used in UTI are shown in Table 1.

In recent years, a small number of new antibiotics have been approved for specific indications. For example, plazomicin is a next-generation aminoglycoside that was derived from sisomicin using a chemical approach that endows activity against resistant bacteria expressing aminoglycoside-modifying enzymes [38]. It was approved by the FDA in 2018 for the treatment of complicated UTIs, including pyelonephritis [39]. In clinical development, studies of plasma and urinary pharmacokinetics demonstrated that urinary concentrations exceed plasma levels after IV administration. A 15 mg/kg single dose delivered mean plasma and 0–4 h post-dosing peak urine drug concentrations of approximately 140 mg/L and 800 mg/L, respectively [40]. Subsequent population pharmacokinetic modelling was employed to simulate the plazomicin concentrations in complicated UTI cases presented by patients with normal renal function or compromised renal impairment [40,41,42]. This study concluded with a simplified and individualised dosing strategy for a spectrum of UTI cases to optimise therapeutic outcome and minimise the risk of adverse effects.

## 4. Factors Affecting Antibiotic Accumulation within the Urinary Tract

The accumulation of an antibiotic in the renal and urinary system is dictated principally by a few key processes—the extent to which it is filtered through the glomerulus, how it is transported across the renal tubules, and its interactions with the urothelium when in the urine.

In the blood, all antibiotics undergo reversible binding to plasma proteins such as albumin. The extent of protein binding varies widely from e.g., 95% for ceftriaxone [43] to 2% for meropenem [44], and is an idiosyncratic feature of each drug determined by its physicochemical properties such as lipophilicity [45]. The fraction of the drug that is not protein-bound, the so-called unbound fraction (f_u_), is the fraction that is available for glomerular filtration. Approximately 20% of the plasma volume passing through the glomerulus at any given moment is filtered to produce primary urine which contains the majority of blood solutes, except most proteins. Thus, for most antibiotics which undergo extensive plasma protein-binding, renal clearance via glomerular filtration plays a minor role in drug clearance. Instead, a balance of active drug secretion into the renal tubules and passive tubular reabsorption are key determinants of renal disposition. For an overview of the role of renal drug transporters, we refer the reader to two excellent reviews [46,47].

Taken together, these data indicate that it is feasible to achieve sufficiently high concentrations in the urinary compartment for many antibiotics. Nonetheless, recurrent, chronic, and severe infection caused by resistant bacteria represent an unmet clinical need that would greatly benefit from the development of innovative technologies for controlled and targeted delivery of antibiotics.

## 5. The Challenge of Recurrent UTI: Biopharmaceutical Challenges to Effective Treatment

Although multiple factors are thought to contribute, recurrent UTIs are considered to be, at least in part, the result of bacterial biofilm production or persistent intracellular bacterial colonies that are recalcitrant to eradication with antibiotics [10,48]. The persistence of intracellular bacterial communities (IBCs) within the urothelial barrier is a challenging predicament. While much of the work on IBCs has been performed in mice, IBCs have also been observed in urothelial cells isolated from women and children with acute UTI [49,50], women with lower urinary tract symptoms [51] and urge incontinence [52], and in biopsies of women with severe recurrent UTI [48]. Additionally, filamentous bacteria, predictive of intracellular bacterial growth, have been observed in urine from women with acute cystitis [53]. Evidence from studies in mice and in vitro cancerous urothelial cell lines indicates that quiescent, biofilm-like architectures can persist for long periods [54]. For antibiotics to effectively clear these intractable communities, additional barriers must be traversed. From a biopharmaceutical perspective, this presents a considerably greater challenge than the treatment of extracellular infections. These challenges are largely underpinned by the necessity of drug crossing the urothelial plasma membrane, a notably formidable barrier, and once there, achieving intracellular concentrations sufficient to achieve bacterial killing. Thus, in the absence of high membrane permeability to the antibiotic of choice, it will be impossible to reach the elevated MICs for these protected bacteria.

The major biopharmaceutical challenge is that the primary function of the urothelium is to provide a barrier against the flux of solutes (including renally-excreted waste products) and pathogens into the systemic circulation. The hypertonic nature of urine (300–900 mOsmolar) presents considerable osmotic and chemical gradients at the epithelial cell surface and highlights the critical protective role played by the urothelium to the maintenance of a safe equilibrium between the blood and urinary compartments. The barrier function of the urothelium to non-specific passage of solutes, particles, and microorganisms is principally presented by a restrictive paracellular pathway. A series of radiolabelled tracer studies on urothelial tissue from different species have demonstrated the permeability to several, small endogenous urinary solutes including water [55,56], urea [57,58], and creatinine [59]. To highlight the selectivity of urothelial solute reabsorption, the clearance of tritiated water in the rabbit and human bladder were 38% in 45 min [58] and 24% in 42 min [56], respectively. The extensive urothelial permeation of water is attributed to the expression of aquaporin channels [60,61] that are widely expressed and serve to regulate water uptake across several mucosal barriers [62]. In the rat there is evidence [63,64] to indicate that expression of aquaporin 2 is increased after bladder distension, although *AQP2* mRNA was undetectable in human bladder [60]. The disappearance of ^14^C labelled creatinine in the rabbit was approximately 5% per hour in the healthy bladder, rising to 22% in the inflamed bladder [59]. The notable increase in the permeability of the inflamed urothelium to water, urea, and creatinine indicates that under such cases of physiological stress, there is likely increased permeation of antibiotic into the urothelium. Permeability to creatinine [59] and xenon [65] was reported to increase markedly in the inflamed urothelium. In contrast, drug permeation into the urothelium is extremely limited [66,67,68,69].

The urothelium is stratified into three distinct cell layers, basal, intermediate, and luminal superficial (umbrella) cells. In humans, these superficial umbrella cells are protected by a crystalline lattice comprising four uroplakins—UPK1A, UPK1B, UPKII, and UPKIII—that adorn up to 90% of the umbrella cell surface [70]. UPK1A and UPK1B are N-glycosylated and are terminated with mannose, which is the ligand for the adhesion protein, FimH, which is expressed on uropathogenic *E. coli* [71] and is responsible for the attachment and subsequent colonisation of the urothelium in many clinical cases of chronic UTI [72].

The tightly-controlled differentiation of the urothelium produces a complex mucosal barrier that protects the underlying cells from the toxic effect of urine, but also poses particular challenges for intracellular drug delivery. The permeability of the urothelium is highly restricted with a demonstrably extremely low flux of electrolytes and non-electrolytes (urea, ammonia).

Paracellular transport in the urothelium is restricted by the apical junctional ring (AJR) complex. The AJR is composed of the tight junctional complex (zonula occludens), the adherens junction, desmosomes, and the associated actomyosin ring. It is responsible for maintaining the integrity of the urothelial surface through cycles of bladder filling, extension, and voiding that is reported to see a greater than 50% increase in the cell perimeter during bladder distension [73]. The tight junctional complex is the uppermost structure, which presents an uninterrupted permeability barrier between neighbouring umbrella cells. This reflects the principal role of the tight junctions in restricting paracellular solute flux from the apical urinary compartment. UTI is associated with increased urothelial permeability due to disruption of the paracellular pathway and breakdown of the tight junction complex [74].

The amalgam of a highly restrictive paracellular and a uroplakin-reinforced transcellular barrier confers the urothelium with an ultra-low permeability profile, which supports its physiological role as an impenetrable liner for the conduits and repository of the urinary tract. From a drug delivery perspective, the challenges for effective antibiotic delivery are distinct for acute and recurrent infections. To treat acute infections, the challenge lies in achieving sufficiently high drug concentrations at the urothelial surface for efficient bactericidal activity. Recent work in the drug delivery field has provided advances in mucoadhesive formulations that could be adapted to antibiotic delivery platforms that provide sustained antibiotic release [75,76,77,78].

The targeting of intracellular bacterial colonies in cases of recurrent UTI presents a considerably greater challenge [79]. To eradicate these pathogens, it is necessary to use an antibiotic with physicochemical properties that favour the transcellular route that directs the drug to/through the cytoplasmic compartment. Blango and Mulvey [80] tested a panel of seventeen antibiotics, spanning different classes, against a reference UPEC isolate cultured as both biofilms and an intracellular infection model using the 5637 human bladder cancer cell line. Of the seventeen tested drugs, only nitrofurantoin and the fluoroquinolones, ciprofloxacin, and sparfloxacin reduced intracellular bacteria. Notably, however, none of the antibiotics were demonstrably active in a mouse UTI model. It is likely that this was due to the fact that the cell culture model used (5637 cells) is unable to reproduce the urothelial barrier function due to not forming AUM plaques.

Recently, Gonzalez et al. examined the extent of dose-dependent intracellular antibacterial activity of ceftriaxone, ciprofloxacin, and azithromycin, using amikacin as a non-cell-penetrant control [81]. They demonstrated potent activity against extracellular infections, but this was less profound in experiments using intracellular bacteria. In these studies, the concentrations of ciprofloxacin deployed were modest due to a significant increase in T24 bladder epithelial cell cytotoxicity at the highest tested concentration of 5 µg/mL. It is noteworthy that the ciprofloxacin concentrations demonstrating bactericidal activity against intracellular bacteria in these studies were measurable in the urine for 42 h (Figure 2). Akin to the 5637 model, the T24 cell line fails to recapitulate the intact urothelial barrier function. The lack of in vivo efficacy against intracellular bacteria indicates that an alternative approach is necessary for the effective clearance of these pathogens. Clinical case reports indicate that intravesical instillation may offer an alternative means of increasing local antibiotic concentrations in the vicinity of the urothelial infection site. Optimal design of an intravesical delivery technology will be dependent on the ability of these agents to efficiently cross the urothelial barrier.

## 6. Intravesical Therapy for UTI

For the most effective treatment, the ideal scenario would be to deliver the right drug, at the right dose, at the right time, to the right target and nowhere else, to maximise safety and efficacy while minimising drug burdens and, hopefully therefore, overall treatment costs. Direct drug-targeting technologies are used extensively to minimise confounding pharmacokinetic variations to provide best quality treatments e.g., eye drop formulations, metered dose inhalers, skin creams/ointments, and intra-tissue injections. Bladder cancer adjuvant chemotherapy and intra-detrusor botulinum toxin injections exemplify the usefulness of direct-to-bladder delivery.

In 2018, Pietropaolo and colleagues [82] published a systematic review on the use and effectiveness of intravesical antibiotic treatment, both for treatment and prevention of UTI. The drugs examined were gentamicin, neomycin/polymyxin, neomycin, and colistin. For studies of treatment (N = 6), an average reduction of 88% was seen (with 5% having discontinued). For studies of prophylaxis (N = 5), an average reduction of 71% was seen (with 8% having discontinued). Discontinuation was more likely with the non-gentamicin group, although overall the review deemed the side effects to be “minor”. The authors concluded that the method was relatively safe and effective for short-term treatment and may therefore serve as a good backup when more traditional antibiotic treatment has failed. In 2022, Reddy and Zimmern also published a systemic review [83]. These authors focused on only three studies using gentamicin and also found that there was an effective reduction in frequency of UTIs; one study showed an 82% decrease; one showed a median of 1.5 UTIs/6 months from 2.5; and one showed a mean of 1.2 from 4.8. Discontinuation rates ranged from 0–22%. Another recent study showed that, in their clinical setting, intravesical gentamicin treatment was generally safe and effective in paediatric urology patients [84]. Finally, Ong and colleagues systematically reviewed the literature on intravesical treatment with aminoglycosides for refractory UTI [85]. Considering 19 publications, 80.7% of patients experienced a successful outcome with the aminoglycoside in question alone, while 79.5% experienced a successful outcome when the aminoglycoside was administered in combination with polymixin. Some 6.2% of patients discontinued, while the microorganisms showed an increased in antimicrobial sensitivity in about 15% of patients on either regimen. Taken together, these studies support the idea that intravesical antibiotic treatment can be safe and efficacious, but as noted by Ong and colleagues, more studies are needed as the sample sizes are relatively small.

Based on this evidence, an intravesical drug-delivery approach shows promise and has generated the interest of patients, clinicians and scientists alike. However, a better delivery system needs to be developed. Drug delivery through standard drainage catheters is fraught with infection risks and handling challenges, especially with patient self-administration. To date, none of the available catheters are certified for this purpose, and pharmaceutical companies will not produce products for delivery through unlicensed devices if there is a licensed alternative treatment. Therefore, an unmet need in the field is for properly designed devices to deliver drugs directly to the urothelium to attempt to prevent and treat UTI.

## 7. Future Advances in Intravesical Therapy

Looking ahead to the future of intravesical treatments for UTI, some have attempted to make antibiotic instillations more efficient through the deployment of nano-sized delivery platforms aiming to permit intracellular drug accumulation via the internalisation of drug-loaded particles through endocytosis (64). For example, Brauner et al. [86] created poly(d,l-lactic-co-glycolic acid) (PLGA) nanoparticles designed to adhere to the bladder wall in the face of dilution from urine issuing from the kidneys, which were then loaded with the antibiotic, trimethoprim. Their in vitro studies in human bladder cell culture showed robust adhesion of the particles despite repeated washing steps, which might extrapolate to longer dwell times of the drug in an intravesical dosing. One of us (JR) has reported [87] nitrofurantoin encapsulated in microparticles in PLGA, which were found to release drug through multiple layers of a stratified human urothelial organ model, killing intracellular bacteria and disrupting biofilms in vitro. In another strategy, we reported [88] ultrasound-activated gas-filled PLGA microbubbles loaded with liposomal gentamicin that exhibited superior delivery and killing of bacteria in the urothelial organoid model.

Given how antimicrobial resistance will only increase globally, however, there have been attempts to move beyond antibiotic instillations in the treatment of UTI. Leitner and colleagues [89], for example, assessed the effect of a commercially available bacteriophage (Pyophage) in a randomised, double-blind, placebo-controlled clinical trial of men with UTI after transurethral resection of the prostate (a total of 97 patients, 28 receiving intravesical Pyophage, 32 receiving intravesical placebo, and 37 receiving systemic antibiotics). The authors found that the bacteriophage treatment was non-inferior to standard-of-care antibiotics with an acceptable safety profile, though they noted that this modality is not yet an approved option.

A number of groups are exploring the use of functionalized antimicrobial particles suitable for intravesical treatment of UTI. For example, in an in vitro cell culture infection model system, Khanal et al. [90] coupled trimeric thiomannoside clusters to biocompatible nanodiamond particles and showed that these could effectively block the FimH adhesin of *E. coli*, which is responsible for host–cell binding and contributes to biofilm formation. Liu and colleagues [91] took a photodynamic therapy approach with their Chlorin e6-encapsulated surface charge-conversion polymeric nanoparticles, which when irradiated by a laser emit reactive oxygen species that can kill bacteria. The particles were effective against *E. coli* and *S. aureus* in an in vitro cell culture infection model, and against *E. coli* in experimentally-infected mice treated intravesically. While promising, the mice needed surgery to introduce the laser treatment, so more refinements in technique would be needed to make this a less invasive option in humans. As a final example, Iyer and colleagues [92] created nanodiamonds that were able to penetrate infected bladder cells in culture and kill intracellular *E. coli*.

## 8. Conclusions

Antibiotics remain the primary treatment for UTI. However, with increasing AMR and in the face of growing evidence of microbiota disruption following systemic antibiotic administration, there is a growing need to better target antibiotic therapies, alongside the development of new non-antibiotic alternatives. In complex cases of UTI, intravesical antibiotic delivery holds the potential to better resolve these infections by enabling the presentation of antibiotic concentrations far in excess of those achievable via the systemic route. By virtue of the negligible urothelial permeability, which prevents drug permeation into the systemic circulation, this route would also logically pose little threat to non-urothelial microbial communities including those residing in the gut which are crucial to overall health. Impediments to the successful exploitation of this route include the availability of licensed administration devices and drug formulations that can be safely and effectively delivered to the urinary tract. Only with appropriate investment in these technologies and the performance of clinical trials in the appropriate patient populations can we properly appraise the potential of intravesical therapy for UTI.

## Figures and Tables

**Figure 1 pathogens-12-00417-f001:**
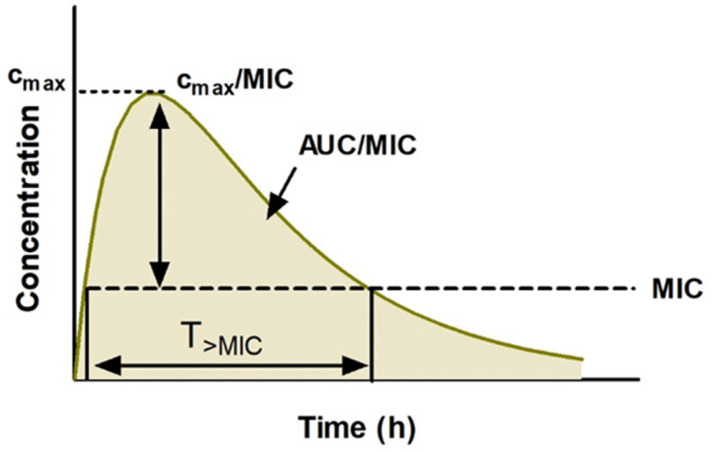
Typical blood plasma antibiotic concentration–time profile after oral administration. The delivered dose is absorbed from the gastrointestinal tract while undergoing simultaneous elimination. Before the concentration peaks (C_max_), absorption proceeds at a faster rate than elimination until the absorbable dose is depleted and the drug undergoes distribution across the body compartments and is eliminated by hepatic metabolism and/or renal excretion. MIC: minimum inhibitory concentration; AUC: area under the plasma concentration–time curve.

**Figure 2 pathogens-12-00417-f002:**
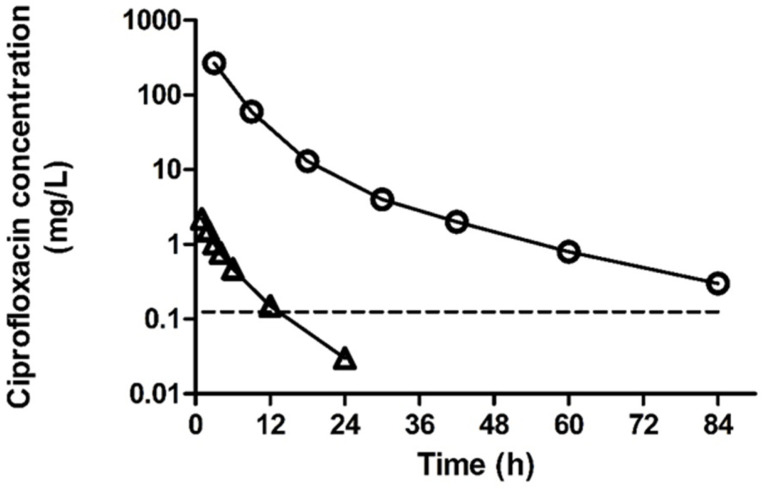
Mean plasma and urine concentrations of ciprofloxacin after a single oral dose of 500 mg ciprofloxacin. ∆: plasma; O: urine. Urinary concentrations are plotted against the middle of the collection time ranges. Data are redrawn from [30].

**Table 1 pathogens-12-00417-t001:** Comparison of peak plasma and urinary concentrations of select antibiotics used to treat UTIs. Adapted from [29].

Antibiotic (Oral Dose)	Approximate Ratio of Urine: Plasma Concentration **	Reference
Amoxicillin (250 mg)	138:1	[31]
Cephalexin (250 mg)	122:1	[32]
Co-trimoxazole	T: 50:1; S: 4:1	[33] *
Nitrofurantoin (100 mg)	50:1	[34] ^†^
Fosfomycin (3000 mg)	105:1	[35]
Ciprofloxacin (250 mg)	148:1	[36]
Levofloxacin (500 mg)	113:1	[37]

* comprising 160 mg trimethoprim (T) and 80 mg sulfamethoxazole (S). ** calculated from the mid-point if a concentration range is shown. ^†^ readers are referred to a comprehensive summary of published urine and plasma PK data for nitrofurantoin.

## Data Availability

No new data were created or analysed in this study. Data sharing is not applicable to this article.

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
