# Peer review of "Effective Treatments of UTI—Is Intravesical Therapy the Future?"

_pathogens, 2023, doi:10.3390/pathogens12030417_

Round 1

Reviewer 1 Report

The comprehensive review manuscript draws attention to an issue of paramount importance, which is the adverse effect of systemic antibiotics on host microbiome, homeostasis and sequalae of chronic diseases. The data driven argument for intravesical administration to treat UTI is timely as we are still battling pandemic, and antimicrobial resistance ensuing from use of systemic antibiotics can complicate the treatment of COVID related complications.

·        Instead of serum concentration, the correlation of urinary concentration of antibiotic with clinical outcomes of UTI patients supports the authors premise for intravesical administration to mitigate antibiotic microbial resistance and metabolic degradation systemic antibiotics with intravesical administration.

  • Water permeability of urothelium discussed on page 7 need to be revised considering significant absorption of tritiated water (40mL/hour) by human patients (PMID: 5083329; 2225866) and in rabbits (PMID: 1688456). Water absorption by urothelium apparently increases with distension mediated upregulation of aquaporin channels (PMID: 29797427; 9042675).
  • The urea permeability of urothelium also need to be updated based on the permeability of radio labelled urea in rabbit bladder (PMID: 1688456) and human bladder (PMID: 6626894).
  • The urothelial permeability is further increased by infection evoked inflammation as demonstrated by the ingress of xenon (PMID: 973287) and creatinine (PMID: 1209795). The increased permeability is relevant in the ingress of antibiotics administered intravesically.
  •       With regard to the risks from antibiotics limiting the protective effect of resident microbiome in bladder, author should discuss (PMID: 29243301).

Author Response

The comprehensive review manuscript draws attention to an issue of paramount importance, which is the adverse effect of systemic antibiotics on host microbiome, homeostasis and sequalae of chronic diseases. The data driven argument for intravesical administration to treat UTI is timely as we are still battling pandemic, and antimicrobial resistance ensuing from use of systemic antibiotics can complicate the treatment of COVID related complications.

  • Instead of serum concentration, the correlation of urinary concentration of antibiotic with clinical outcomes of UTI patients supports the authors premise for intravesical administration to mitigate antibiotic microbial resistance and metabolic degradation systemic antibiotics with intravesical administration.

We wish to thank this reviewer for their warm support for our article, especially the indication of its timeliness.

  • Water permeability of urothelium discussed on page 7 need to be revised considering significant absorption of tritiated water (40mL/hour) by human patients (PMID: 5083329; 2225866) and in rabbits (PMID: 1688456). Water absorption by urothelium apparently increases with distension mediated upregulation of aquaporin channels (PMID: 29797427; 9042675).
  • The urea permeability of urothelium also need to be updated based on the permeability of radio labelled urea in rabbit bladder (PMID: 1688456) and human bladder (PMID: 6626894).
  • The urothelial permeability is further increased by infection evoked inflammation as demonstrated by the ingress of xenon (PMID: 973287) and creatinine (PMID: 1209795). The increased permeability is relevant in the ingress of antibiotics administered intravesically.

We have extended our discussion of the extant data pertaining to solute clearance kinetics (water, urea, creatinine) from the bladder lumen and expanded our commentary on the relative impermeability of the urothelium. These can be found in lines 243-260 in the updated manuscript.

With regard to the risks from antibiotics limiting the protective effect of resident microbiome in bladder, author should discuss (PMID: 29243301)

Thank you for this comment; we agree it is an excellent idea to mention studies that support the indirect evidence for the protective action of the urobiome. However, as ample human studies exist, instead of mentioning a small study in nine dogs, we thought it would be more effective to cite a review summarising the seven clinical trials (including RCTs) of bladder instillation of the ASB strain E. coli 83972 in recurrent UTI patients. This has been added in lines 73-6 of the updated manuscript.

Reviewer 2 Report

This work is a comprensive review of UTI treatments. I honestly think that authors explained correctly and in a very logical and understandable way the problematic and the hypothetical solution with intravesical antibiotic delivery. 

Author Response

This work is a comprensive review of UTI treatments. I honestly think that authors explained correctly and in a very logical and understandable way the problematic and the hypothetical solution with intravesical antibiotic delivery.

We thank the reviewer for their glowing review of our article. I hope they will appreciate the improvements that have been made in response to other comments.

Reviewer 3 Report

I would like to thank the authors for a very interesting work on a current topic with a lot of information about biopharmacinetics, 

I had only a few comments:

- Even if it is a review article, after  it is needed to structure the manuscript: Introduction&objectives, mat&methods, results, discussion, conclusion. In the manuscript , many of these are lacking.

-The introduction part, or first part written about antibiotics is very long and in the end the topic of the article (intravesical treatment for UTI) is very reduced and almost lost at the end of the manuscript with just a brief comments of other articles. Tables of antibiotic success rate or reduction of UTI with complete statistic parameters (p value included) might be useful to make this part a bit more consistent.

- It reads as a narrative review but it is unclear how the search was conducted and it also combines a first part which is more as a antibiotic pharmacocinetic explanation in the line of text book. Important to understand the topic, but maybe too extensive

- Important to add a discussion part, where to put the focus of the title: Effective treatments of UTI – is intravesical therapy the future?

Success rate/ failure /availability problems etc---

Author Response

  • Even if it is a review article, after  it is needed to structure the manuscript: Introduction&objectives, mat&methods, results, discussion, conclusion. In the manuscript , many of these are lacking.

We have prepared the manuscript in strict accordance with the editorial instructions from the Journal.

-The introduction part, or first part written about antibiotics is very long and in the end the topic of the article (intravesical treatment for UTI) is very reduced and almost lost at the end of the manuscript with just a brief comments of other articles. Tables of antibiotic success rate or reduction of UTI with complete statistic parameters (p value included) might be useful to make this part a bit more consistent.

Our discussion of intravesical antibiotic efficacy aims not to cover in depth the systematic reviews that have been published by others (e.g. Pietropaolo et al.) for different patient populations. Instead, we aimed to give a broad overview of the evidence for the safe and effective use of the intravesical route for antibiotics.

  • It reads as a narrative review but it is unclear how the search was conducted and it also combines a first part which is more as a antibiotic pharmacocinetic explanation in the line of text book. Important to understand the topic, but maybe too extensive

We sought to provide a comprehensive overview of the key antibiotics used for UTIs and to highlight their dispositional challenges. We believe this structured approach allows the reader to follow our arguments and form their own opinion of the clinical opportunities for intravesical delivery. Furthermore, we received no such criticism from the other referees.

- Important to add a discussion part, where to put the focus of the title: Effective treatments of UTI – is intravesical therapy the future? Success rate/ failure /availability problems etc---

A discussion was included, although due to journal guidelines, it was not labelled as such. We have provided a succinct commentary on the clinical evidence (including success rates) for intravesical therapy and a future perspectives section on lines 321 - 401. Lines 412 onwards highlight the impediments to successful exploitation of this route and indicate key areas for future investigation.

Round 2

Reviewer 3 Report

Thank you for your response.